# Visual Attention and Recognition Differences Based on Expertise in a Map Reading and Memorability Study

**Merve Keskin** [1,*] , **Vassilios Krassanakis** [2] **and Arzu Çöltekin** [3]

1   Finnish Geospatial Research Institute, National Land Survey of Finland (FGI/NLS), Vuorimiehentie 5, 02150 Espoo, Finland

2   Department of Surveying and Geoinformatics Engineering, Egaleo Park Campus, University of West Attica, Ag. Spyridonos Str., 12243 Egaleo, Greece

3   Institute of Interactive Technologies, School of Engineering, University of Applied Sciences and Arts Northwestern Switzerland, Bahnhofstrasse 6, 5210 Windisch, Switzerland

*   Correspondence: merve.keskin@nls.fi

**Abstract:** This study investigates how expert and novice map users' attention is influenced by the map design characteristics of 2D web maps by building and sharing a framework to analyze large volumes of eye tracking data. Our goal is to respond to the following research questions: (i) which map landmarks are easily remembered? (memorability), (ii) how are task difficulty and recognition performance associated? (task difficulty), and (iii) how do experts and novices differ in terms of recognition performance? (expertise). In this context, we developed an automated area-of-interest (AOI) analysis framework to evaluate participants' fixation durations, and to assess the influence of linear and polygonal map features on spatial memory. Our results demonstrate task-relevant attention patterns by all participants, and better selective attention allocation by experts. However, overall, we observe that task type and map feature type mattered more than expertise when remembering the map content. Predominantly polygonal map features such as hydrographic areas and road junctions serve as attentive features in terms of map reading and memorability. We make our dataset entitled CartoGAZE publicly available.

**Keywords:** eye tracking; AOI; spatial memory; memorability; cartographic usability; task difficulty; expertise; eye tracking dataset; navigational maps

## 1. Introduction and Background

Maps are essential to our everyday lives. In addition to navigational or way-finding tasks, we use them as visual aids to learn about phenomena varying from statistical distributions through landscapes to examine how our spatial surroundings are organized. While some maps are highly specialized, most commonly used maps are those that are served online and designed for a general audience, such as Google Maps, Apple Maps, and OpenStreetMap.

Whether they are designed for the general public or specialized user groups, a fundamental way to think about map design is through *visual variables*, such as size, position, shape, orientation, texture, color hue, and value (lightness or intensity) [1]. These primary visual variables are typically modifiable and can be found on every map (or any other visuospatial display), and they can be used to guide the design process and help characterize map elements. How these visual variables are designed, and thus how the content is distributed within the map drawing area, can have a great impact on human spatial cognition when working with maps [2]. In addition to these modifiable visual variables, for a map of the environment (as opposed to, e.g., a thematic map), the main structuring elements of maps such as main roads, major green areas, and hydrographic features act as visual landmarks around which map readers orient themselves and acquire spatial knowledge [3]. We call these main structuring map elements 'map landmarks'.

Eye movements provide insight on cognitive procedures involved in spatial knowledge acquisition, i.e., in forming mental representations and the hierarchical organization of map landmarks in spatial thinking processes [4–7]. However, spatial knowledge acquisition from maps is far from fully understood, and more research is needed on questions such as the following:

- Which map landmarks are easily remembered? (**memorability**)
- How are task difficulty and recognition (or cued recall) performance associated? (**task difficulty**)
- How do experts and novices differ in terms of recognition performance? (**expertise**)

This paper is primarily motivated by fundamental science questions—reproducing perceptual psychology results with maps to better understand human geospatial information processing—rather than applied questions. Though we see some everyday relevance to our study eventually, e.g., when driving or walking, one does not have to look at the map as often if the maps are designed to boost memorability effective recognition of relevant features, with knowledge gained in studies such as ours. Furthermore, understanding the visual characteristics of what is memorable and easier to recognize in the context of map use may potentially serve educational and socio-political communications better. This is why examining the link between recognition, memorability and expertise explored via visuospatial tasks would have both fundamental and eventual applied science relevance. Given the context above, below we briefly review the related work on **memorability**, **cued recall and task difficulty** and **expertise on spatial memory** in perceptual psychology, cognition and geospatial information science domains.

*Memorability, recognition and recall.* In connection to our first research question "which map landmarks are easily remembered", we draw some parallels to existing research on image and visualization memorability. An important question in this corner of research is perhaps "if an image is difficult to recognize, is it also difficult to recall?" [8]. Despite the variation in individual experiences and characteristics, studies in psychology and neuroscience show that it is possible to quantify the memorability of an image and predict it for a reasonable number of cases [9]. For image recollection tasks, as measured by memory-based drawings, Bainbridge et al. [10] found that visual saliency and "meaning maps" can explain aspects of memory performance, but they observed no relationship between recall and recognition of individual images. This observation is somewhat contradictory to research outcomes in nameability, where it has been demonstrated that if visual features such as colors are nameable, they are also easier to remember [11]. Given that the link between recall and recognition is not fully established, then what makes a visualization memorable? As Borkin et al. [12] stated "In human cognition, understanding and memorability are intertwined", memorability should include both recognition and recall, and ideally lead to long term retention. Borkin et al. [12] exactly examine how visualizations would be remembered if they were images, and demonstrate that visualizations containing pictograms, more color, low data-to-ink ratios, high visual densities, and those that contain novel and unexpected features are of higher memorability than 'clean visualizations' and the ones with limited variability [12]. Borkin et al.'s [12] findings broadly agree with, e.g., Lenneberg's [11] findings about color nameability. However, it is important to note that Borkin et al.'s work is at a different granularity level (as it deals with visualizations) than the other studies we cite here where the focus is more on natural scene gist memory [13]. Most maps are closer to visualizations than natural scenes, and when it comes to maps, it has been shown that the presence of visual landmarks on a map can improve route recall [14]. More specifically, eye tracking studies have shown that linear map features such as roads or rivers were easier to learn and remember, even though the viewers did not pay much explicit attention to such features (e.g., [15–17]). Furthermore, map landmarks such as topographic details and grid lines appear to guide viewers' attention toward to-be-learned object locations, improving memory performance [18,19].

*Cued recall and task difficulty.* Our second research question on task difficulty vs. cued recall performance is mostly informed by the literature on working memory (WM).

WM refers to the systems and processes involved in the control, regulation, and active maintenance of task-relevant knowledge in the service of complex cognition during both novel and familiar tasks [20]. As Chai et al. [21] stated, WM is heavily engaged in goal-directed activities in which information must be kept and modified for the task to be completed successfully. Cognitive load is closely linked to working memory capacity, and overusing WM is mostly a result of attention devoted to the task demands. Therefore, success at performing a task is determined by the cognitive resources allocated for that task and how long this allocation lasts [22–24]. Recent studies on visuospatial recall tasks controlling for visual complexity and individual differences point out the impact of WM capacity and task difficulty on the rate of obtained spatial knowledge (e.g., [25–27]). For example, Lokka and Çöltekin [27] studied the visuospatial memory capacity on route learning in virtual environments and found that although a reduced visual complexity improved recall accuracy, success rates were specifically low (around 60%) in a difficult task that required switching perspectives. Findings such as these suggest that manipulating visual design can only do so much when the tasks require high levels of cognitive processing. This is in line with the findings from other disciplines, in fact, there are well-established links between visuospatial task complexity and cognitive load. In studies that focus on measuring working memory capacity or cognitive load, it has been shown that split attention (i.e., dual-task scenarios), and the complexity of the phenomena (e.g., words, routes, shapes) impair recall performance [28–30].

*Expertise.* While task difficulty is important, a person's previous exposure to similar tasks (i.e., their expertise) might moderate how difficult a task is for an individual. Our last research question is related to the influence of expertise on (spatial) memory, thus here we briefly present the related work. The role of expertise in task performance has been a subject of interest in many disciplines with little exchange between them [31]. Brams et al. [32] provide a systematic review on the role of expertise based on three theoretical accounts: the long-term working memory theory [33], the information-reduction hypothesis [34], and the holistic model of image perception [35]. These theoretical frameworks describe underlying processes of how expert performance is influenced by perceptual-cognitive skills, and Brams et al. [32] examine the validity of each based on gaze features. For instance, in highly efficient searches, attention is guided to the target item and the rest of the scene becomes irrelevant. This is called *the guided search model* [36] and forms the basis of the information-reduction hypothesis. In this case, it is expected that the viewer has longer fixation durations and more number of fixations on task-relevant areas and vice versa on irrelevant areas, resulting in high selective attention allocation [37]. Perceptual-cognitive tasks can be completed more efficiently by allocating selective attention. Although no one theory fits all domains of expertise, Brams et al. [32] conclude that experts can maximize their attention to relevant (visuospatial) information, and optimize performance in specific perceptual-cognitive tasks. In general it is understood that expertise matters despite the limited evidence on individual differences in cognitive processes while executing map-related tasks (e.g., [38,39]). In an early example, Thorndyke and Stasz [40] examined expert and novice abilities to learn and recall the information presented via typical planimetric maps and found no significant difference between them. In another study, finding modest differences between experts and novices, Kulhavy et al. [41] speculated that the general map knowledge that the novices obtain from everyday exposure may explain why their performance is comparable to that of experts [41]. Based on the existing eye tracking literature, we know that experts have better defined eye-scanning patterns, mostly have shorter reaction times and fixations, more fixations per second (e.g., [17,42–44]), and fewer saccades (e.g., [45]), all of which are presumably correlated with a lower cognitive load in experts compared to non-experts. Similarly, in a study where complex soil maps with many categories of soil were represented, people with higher self-reported expertise levels overall outperformed those who did not, whereas legend design and task difficulty moderated or even reversed these outcomes [46]. However, in some studies, no significant differences were observed between the expert and novice groups [47,48]. For example, Keskin et al. [5]

observed no significant difference between experts and novices in spatial memory tasks, possibly due to a ceiling effect (high success rates in both groups), where participants worked with the familiar Google navigational maps (i.e., the traditional cartographic maps that many people use every day).

*Our Contributions*

It is well understood that as much as task and expertise, map design characteristics play an influential role on users' cognitive processes and learning performance, hence, assessing them contributes to enhancing the design and usability of cartographic products. More specifically, obtaining detailed insights into the gaze behavior of map users contributes to steering map design toward user-centered thinking, resulting in more usable and useful maps. This study aims to ascertain how expert and novice map users' attention is differently influenced by the map design characteristics of 2D web maps based on an original methodological framework to analyze large volumes of eye tracking data. Different from the previous work, we respond to the research questions mentioned earlier based on an automated area-of-interest (AOI) analysis. The automated AOI approach constitutes a script-based batch processing approach that considers multiple variables, i.e., the vector characteristics, visual variables of the map features, task difficulty, expertise, and spatial memory strategies of human operators.

Furthermore, we distribute our rich eye movement dataset with this publication which provides some value for methodological research on eye movement analysis besides what is presented in this publication. Most open eye movement datasets (e.g., [49]) include data from "free viewing" conditions in the sense that participants usually do not perform specific tasks. These datasets can be used in saliency analyses (e.g., [50]), but do not provide information for task-driven cognitive assessments. We share a dataset containing large eye movement data together with the map stimuli, the AOI files, the task descriptions, and full procedural details for the reproducibility of results and to create possibilities for future research. In addition to publishing the analyses of the empirical data from a controlled experiment, opening our dataset and providing the analysis protocols are original contributions of this publication. The free distribution of the collected dataset to the scientific community could (i) provide the opportunity for other scientists to extend our research results with further analysis as well as to examine new methodological approaches (e.g., by including new indices and/or aggregated visualization methods); (ii) help the process of computational modeling of visual attention for this specific type of visual stimuli (i.e., maps); and (iii) support studies related to artificial intelligence (AI) applications.

## 2. Methodology

### 2.1. Experiment Design

To investigate the above-mentioned research questions on memorability, task difficulty, and expertise, based on the insights obtained in the previous research and empirical evidence found in the relevant literature, we formulated the following hypotheses:

- **H1: Expertise x task difficulty:** Experts might spend more time and have longer fixation durations for task-relevant features than novices do due to their ability to identify what is task-relevant, as well as possibly higher motivation to complete the given tasks.
- **H2: Task difficulty x expertise**: Task difficulty might moderate the effects observed in H1 due to increased cognitive load, especially for non-expert users.
- **H3: Map feature type x expertise (1):** Map features that are complex (e.g., road junctions) and large in size albeit simple (e.g., green areas and hydrographic areas) might draw more attention, and consequently be more memorable than moderately complex features due to a known coupling between attention and memorability. We expect this effect to be more pronounced for expert users as they might be more driven (reasoning similar to H1).

- **H4: Map feature type x expertise (2):** Attention and memory differences between experts and novices might be more pronounced in polygon map features since previous work has shown that linear features are easier to learn and remember irrespective of expertise.

To test these hypotheses, we designed a mixed factorial randomized block design experiment where the following are our **independent variables**: Five map features (all of them based on screenshots of Google's 'navigational maps'), seven task types representing linear and polygon features within the experiment blocks, and two expertise levels (i.e., experts vs. novices). The tasks are grouped into three difficulty levels based on a qualitative assessment and pilot studies; easy, moderate, and hard (Section 2.4). Figure 1 depicts the overview of our experiment design and the main independent variables. As **dependent variables**, we measured reaction time, and success rate (accuracy) (Section 3.3), and analyzed fixation durations of collected eye movement data. We controlled for the age and gender of the participants of both groups, screen size, and the number of experimental blocks.

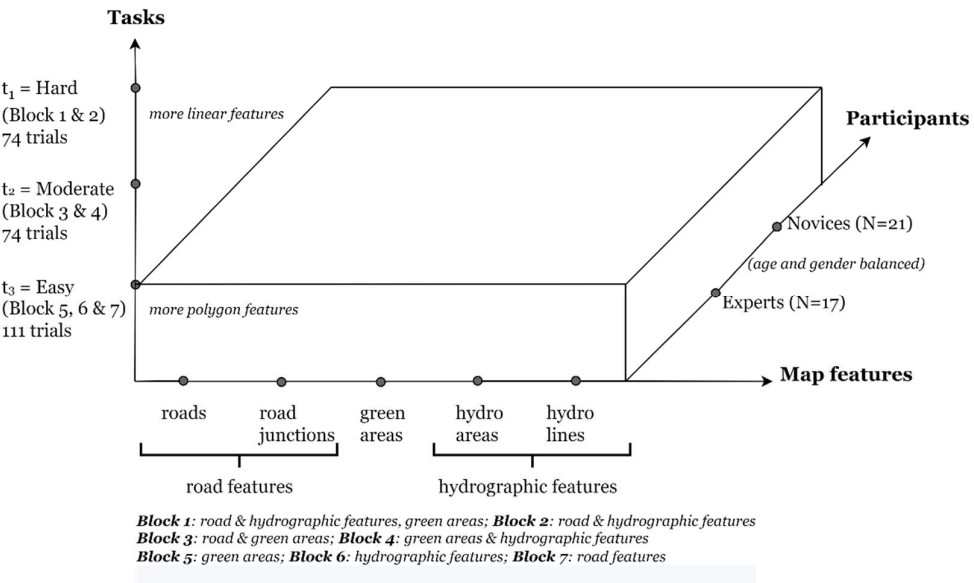

**Figure 1.** Overview of the experiment design. Blocks are further explained in Section 2.4.

In the following sections, we first further detail our experiment (apparatus, participants, task, stimuli, and procedures) and explain the novel analysis approaches we explored.

### 2.2. Apparatus

Participants' eye movements were recorded with a non-contact SMI RED250 eye tracker mounted to the stimulus monitor. The stimulus was shown on a 22" color monitor with 1680 × 1050 px resolution. Participants were placed at 70 cm from the screen with a chin rest to prevent head movements and maintain a fixed distance. The horizontal and vertical eye positions for both eyes were recorded at a frequency rate of 250 Hz with a gaze position accuracy of 0.4° and a spatial resolution (root mean square) of 0.03°.

### 2.3. Participants

This research is approved by the Ethics Committee of the Faculty of Business and Economics of Ghent University where the eye tracking experiments were executed within the framework of the doctoral research project entitled "Exploring the Cognitive processes of Map Users Employing Eye Tracking and EEG" [48]. We used the eye tracking data of 38 participants (*M*age = 29.6, SD = 4.9, *range* = 23–32) of which 21 novices (9 females, 12 males) and 17 experts (9 females, 8 males). We considered participants who hold at least an MSc degree in geomatics and other geospatial information sciences experts. The novices

were volunteers who had no professional experience with maps. We asked how often our participants use Google maps, and to rank the maps for usability (1–5 scale). An analysis of the map use frequency vs. performance is provided in Appendix A. Majority of the participants reported that they use Google maps every day or once or twice a week, and find them user-friendly. Participants reported that they have normal or corrected-to-normal vision, and no one reported having color blindness.

### 2.4. Stimuli and Tasks

The original map stimuli were acquired from Google maps at zoom level 15 with a 1 km scale bar. We followed Google's guidelines while preparing the figures. Since the resolution of a map with the Web-Mercator (EPSG: 3857) projection depends on the latitude and our screenshots are collected from regions all around the world, the scale of the maps varies slightly, from an average scale of 1:40,000. In this study, we work with globally distributed 37 locations as screenshots obtained from Google navigational maps (i.e., their conventional cartographic maps). All screenshots are of globally similar visual complexity levels and all of them contain linear and areal features, i.e., our main map landmarks that we examine in the experiment.

Once the screenshots were obtained as stimuli (Figure 2a), we prepared simplified visual representations of these showing only the task-relevant map landmarks and called them *skeleton maps* (Figure 2b). Participants were asked to study stimuli maps for seven seconds such as shown in Figure 2a in the encoding (learning) phase, which was then removed from their view, and they were provided four skeleton maps (Figure 2b) in the decoding (recognition) phase and asked to mark which one matches the map they have just seen. Preparing the distractor options shown in Figure 2b was important. If multiple map features were to be remembered (e.g., roads and hydrography), a participant might remember only one type (e.g., hydrography), and then find a correct skeleton map based only on this type of information. Thus, the options in the graphical answer screen assured that a response based on partial information was impossible.

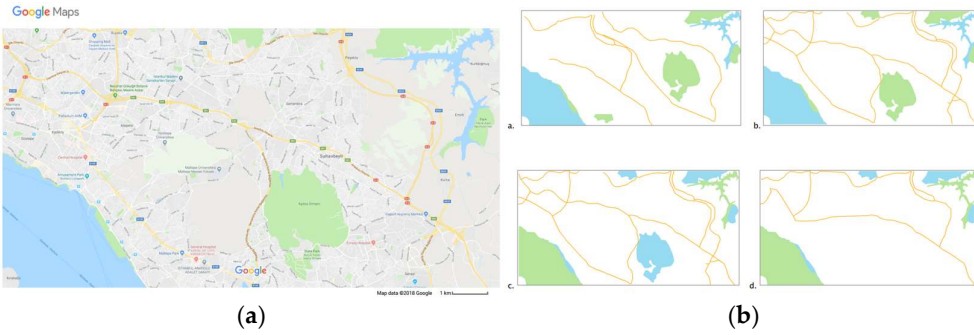

(**a**)　　　　　　　　　　　　　　　　　　　　(**b**)

**Figure 2.** (**a**) Example stimulus ((**a**) larger version of the map(s) can be seen in Appendix B); (**b**) correct skeleton map in multiple-choice graphical answer screen (right).

We generated the skeleton maps by digitizing all the main roads, road junctions, major hydrographic features, and green areas on the original map stimuli using GIS software. These digitized map landmarks also correspond to the AOIs created (elaborated in Sections 2.6 and 2.7). In total, 1036 skeleton maps (i.e., *37 trials × 7 blocks × 4 graphical options*) were used in the graphical response screens. Stimuli used in this study are a subset of those provided by Keskin et al. [51].

As mentioned earlier, we designed seven experimental blocks which are essentially our *task types* in three difficulty levels (i.e., hard, moderate, and easy). We qualitatively judged the difficulty levels based on initial experiments within the research team and pilot studies, as well as based on the literature on visual complexity and memorability. For the pilot tests executed with ten participants before the main experiment, we calculated inverse efficiency scores (i.e., reaction times of correct answers) and observed clustering of certain

blocks and natural breaks between Blocks 2 and 3 and Blocks 4 and 5. Consequently, we deem Blocks 1 and 2 as *hard*, Blocks 3 and 4 *moderate*, and Blocks 5, 6, and 7 *easy*. This was confirmed by the inverse efficiency scores calculated for all 38 participants after the main experiment (for detailed information [51]).

All tasks used in this experiment are visuospatial memory tasks; participants were required to remember either linear or polygon (areal) map features or combinations of them. The difficulty recognizing (and thus, memorability) of a single feature may be quite different to remembering all features. We selected some of these tasks to see if some of the arguments from perceptual psychology experiments (i.e., very controlled, simplified stimuli) may also work with maps (i.e., a lot more contextual information). We purposefully do not have a concrete end-user scenario, so the questions are more at the fundamental science level than at the applied science level. Ecological validity might often be at odds with experimental control but avoiding a concrete single-use case helps with generalizability of the results.

In this experiment, three main map feature classes were our interest: roads, hydrography, and green areas. *Roads* refer to the main roads (linear) and road junctions (mainly polygon, i.e., triangles, roundabouts, two or more crossing lines), and *hydrographic features* are the main rivers/streams (linear), and major water bodies (polygon). *Green areas*, on the other hand, were all polygons. Trials in Block 1 were designed to study participants' recognition performance of the map landmarks across the entire map, thus was the hardest since participants had to process all of the information. The trials in Blocks 2, 3, and 4 were dedicated to the retrieval of the combination of several map feature classes: Block 2 was very similar to Block 1 in the sense that it included combinations of the 'harder' features, i.e., road and hydrographic (linear and polygon) features. Block 3 included road features and green areas, and Block 4 hydrographic linear and areal features and green areas. The trials in Blocks 5, 6, and 7 dealt with a single map feature class; either green areas, hydrographic or road features, respectively.

There existed a definite number of map landmarks in every map stimulus; either linear or polygon. However, each block used different combinations of them in the multiple-choice graphical answer screens. For instance, let us assume that there are 50 landmarks in one map stimulus (i.e., 10 roads, 10 road junctions, 10 green areas, 10 water bodies, and 10 rivers). In this case, Block 1 would include all 50 landmarks and the rest of the blocks would be as follows:

Block 2: roads + road junctions + water bodies + rivers = 40 landmarks,
Block 3: roads + road junctions + green areas = 30 landmarks,
Block 4: green areas + water bodies + rivers = 30 landmarks,
Block 5: green areas = 10 landmarks,
Block 6: water bodies + rivers = 20 landmarks, and
Block 7: road + road junctions = 20 landmarks.

Since the tasks in each block demanded the recognition (from the memory) of different linear or polygon map features (or a combination of those), the skeleton maps shown in the multiple-choice graphical answer screen were varied. Overall, each block consisted of 37 map stimuli—37 trials (i.e., one for each stimulus), and 1036 skeleton maps to be shown on the answer screen.

*2.5. Procedure*

After welcoming the participants, we first asked them to sign an informed consent form and calibrated the eye tracker. Trials included an encoding (learning) stage and a decoding (recognition) stage. By 'encoding', we refer to the process of how information enters into memory from sensory input (i.e., converting information in working memory to knowledge in long-term memory). We use 'decoding' for the retrieval of previously encoded information. Therefore, it involves processes to access information stored in long-term memory and bring it into working memory. In the encoding stage, we asked the participants to study a map stimulus for seven seconds and told them that they should memorize the map landmarks relevant for the current task type in that block (i.e., indi-

vidual features or combinations of green areas, water bodies, major rivers and roads, and road junctions). After seven seconds, the map stimulus automatically disappeared, and thus at the decoding stage, participants were asked to indicate the correct skeleton map corresponding to the stimulus they had just studied out of four graphical options presented to them from the memory where only one answer was correct (Figure 3).

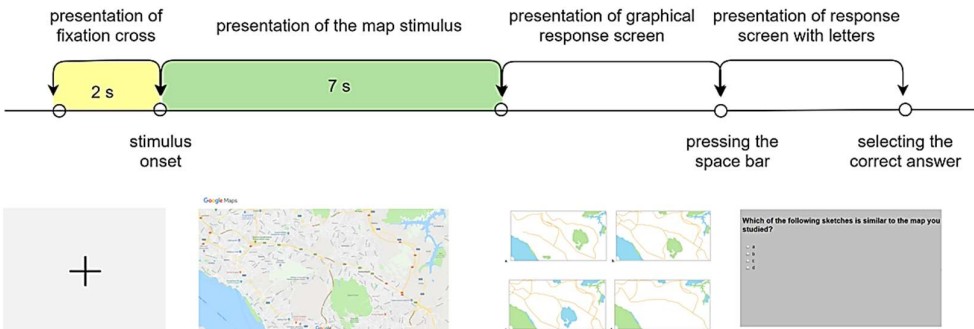

**Figure 3.** Experiment procedure.

It is important to note that each trial started with a presentation of a fixation cross for two seconds long (i.e., 37 fixations crosses and 37 trials per block), this was to be able to average and distinguish EEG signals for reference and activation period (EEG analyses not included in this paper, see [5]). To find the optimum trial duration (i.e., seven seconds), we ran several pilot tests under the supervision of two experimental psychologists, and also paid attention that one block does not last longer than 30 min to avoid fatigue. Additionally, the effect of fatigue is minimized by allowing participants to have small breaks between experimental blocks. The experiment automatically paused at the end of every block and participants were free to move, talk and start the next session at their own pace. The eye tracker was recalibrated at the beginning of every block. The experiment was rather long compared to a standard eye tracking experiment as it was originally designed for eye tracking-EEG coregistration [48], and in a typical EEG experiment, 50–100 trials exist per condition. In this paper, we analyze 37 trials per block and a total of 259 trials over seven blocks. After the main trials, participants rated the task difficulty subjectively, which we also report in this paper. Other data collected in the main experiment, including EEG measures, are beyond the scope of this paper.

### 2.6. Creating AOIs for Map Landmarks

Commonly used in eye tracking studies, marking AOIs around the objects of interest allows precise top-down hypothesis-driven analysis of attention distribution of the participants. The AOIs in this study correspond to map landmarks that participants were required to remember in the experiment. The AOIs for green areas, water bodies, and road junctions were drawn manually on all map stimuli using the open-source GIS software QGIS. We treated the intertwining AOIs (e.g., polygon in polygon, line in polygon, and line intersecting line) separately for more rigorous analyses. For the AOIs of major rivers and main roads, 5 m buffers were drawn considering the precision of gaze position and pixel accuracy depending on the eye tracking device (Figure 4).

Consequently, we created AOIs for all 37 map stimuli used in this experiment and generated a total of 27,023 AOIs of all map landmarks, specifically, 8948 green areas, 4034 water bodies, 3830 rivers, 9220 roads, and 991 road junctions. The overall area coverage of AOIs in all stimuli (100%) is as follows: 5.1% for green areas; 2.7% for water bodies; 2.0% for rivers; 4.2% for roads, 0.3% for road junctions (i.e., *within-AOI*), and 85.9% task-irrelevant areas (i.e., *outside-AOI*) (Figure 5).

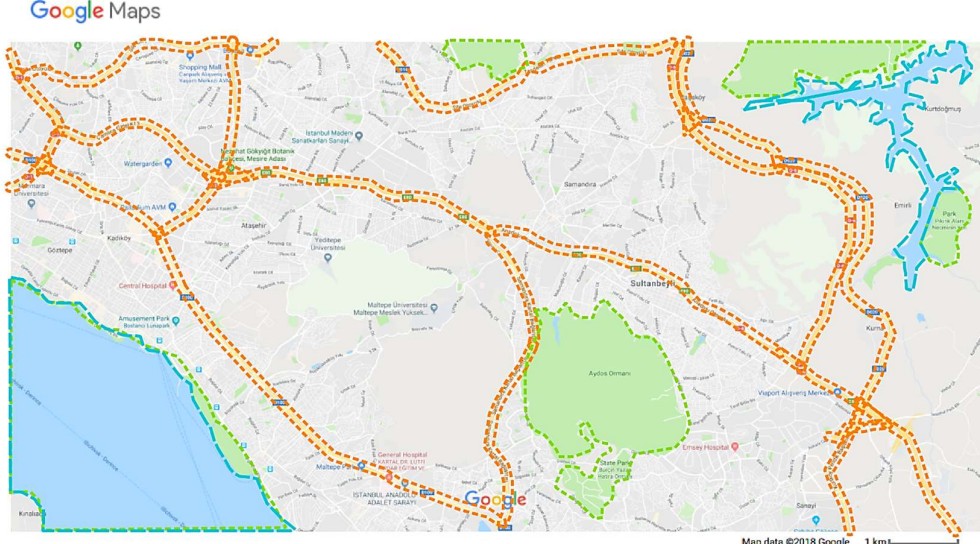

**Figure 4.** Example stimulus and corresponding AOIs (marked with dashed lines) a larger version of the map(s) can be seen in Appendix B.

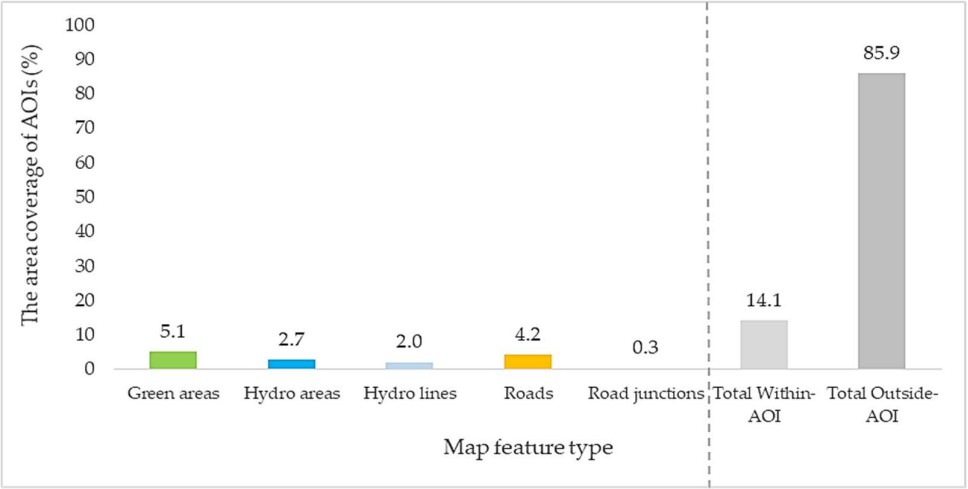

**Figure 5.** The area coverage of AOIs (%) based on the map feature type. Together, within- and outside-AOIs cover the entire screen (100%) as shown on the right-most two bars.

The coordinates of all AOIs were extracted after developing and executing a geospatial model using Graphic Modeler (QGIS) and finally transformed into map image coordinates to perform AOI analysis. Not only due to the large size of the eye tracking data used in this study, but also the large amount of AOIs that we created, we developed a methodology to automate the process of AOI-based fixation analysis, which we explain in the next section (Section 2.7).

### 2.7. AOI-Based Fixation Analysis of Map Landmarks

#### 2.7.1. Encoding Stage

The main AOI analysis of the collected eye movement data was conducted for the encoding stage when the participants studied the map stimuli. We chose average fixation duration as a metric for this analysis because fixation duration is a measure of attention, and thus it represents the level of engagement with the contents of a visual stimulus, and we work with the premise that attention and memory are linked. Examining participants' fixations on an AOI contributes to the understanding of task-relevant visual behavior, and can help determine the influence of the characteristics of map elements on attention.

Therefore, we calculated fixation durations *within-AOIs* (task-relevant areas) in each block considering the five map feature types; roads, road junctions, green areas, rivers, and water bodies. We also calculated average fixation durations for regions *outside-AOIs* (task-irrelevant areas which correspond to the total screen area except AOIs) and compared "within- vs. outside AOI" to have an explicit analysis of the proportional time spent on task-relevant features. In that vein, we computed the fixations based on the implementation of the dispersion-based (I-DT) algorithm, imported in both EyeMMV [52] and LandRate [53] toolboxes. The I-DT algorithm has already been successfully implemented in large-scale eye tracking datasets [54,55]. The algorithm considers both spatial and temporal constraints to identify fixation events in eye tracking data. Spatial constraints are applied using a two-step spatial dispersion threshold; the first one is the typical spatial threshold implemented in I-DT algorithms, and the second spatial threshold is used to remove the spatial noise of the eye tracking equipment when it is known or when it can be measured (see, e.g., the work provided by Ooms and Krassanakis [56]). However, the same algorithm can also be used in a one-step identification process by considering the same value ($t_1 = t_2$) of the threshold for the two corresponding spatial parameters (see, e.g., the work presented by Krassanakis [53]). Moreover, the minimum fixation duration threshold is implemented to fulfill the temporal constraints during the fixation events identification process.

We manipulated all raw eye tracking data in the pixel coordinate system for spatial dimension, and in milliseconds (ms) for the temporal dimension. Hence, the spatial threshold was imported in pixels (43 pixels) based on the average distance between the subject and stimuli monitor (70 cm), the resolution (1680 × 1050 px), and the physical dimensions (474 × 297 mm) of the stimuli monitor and corresponds to 1o of visual angle. We set the minimum fixation duration to 80 ms as recommended by Popelka [57] and Ooms, and Krassanakis [56], and as it is a finer granularity than the 100 ms recommended by Manor and Gordon [58] for visual and cognitive studies. The practical implementation of the fixation analysis process was completed in MATLAB (MathWorks $^\circledR$) and based on the modification of the original source code provided by Krassanakis et al. [52] on GitHub (https://github.com/krasvas/EyeMMV, URL (accessed on 10 January 2023). To consider only the spatiotemporal coordinates of the eye tracking data collected during our experiment, we wrote and executed a dedicated script in Python. We also developed a dedicated script in MATLAB to extract fixation events that occurred within each examined AOI.

### 2.7.2. The Encoding vs. Decoding Stage

This eye tracking experiment was originally designed to identify and quantify the visual behavior and attention patterns of map users during the encoding stage. Previously, we observed no significant difference between experts and novices while studying map stimuli for similar spatial memory tasks [5,16,51], and as a post hoc research question, in this paper, we briefly explore whether the eye movements suggest an overall difference between the encoding and decoding stages. Specifically, we performed a global post hoc analysis for the entire stimuli (i.e., *not* AOI-based) during the decoding stage, where participants retrieved the information held in the encoding stage and made their choice on the graphical answer screen where all four options were displayed at the same time. This is a preliminary qualitative investigation. In support of these preliminary observations, we also provide a visual comparison of the gaze density maps ('heatmaps') and scan paths of experts and novices for the two phases of Block 1 where participants examined the stimuli as a whole (i.e., global recognition task containing all map landmarks). These exploratory analyses trigger new hypotheses for our next systematic analysis.

## 3. Results

### 3.1. Encoding Stage

Since the AOI sizes for map feature types are not equal (Figure 5), we normalized the fixation durations using inverse weights calculated based on the area coverage of AOIs for

each map feature type, i.e., *17.0 for green areas; 32.3 for hydrographic areas; 43.9 for hydrographic lines; 20.5 for roads, and 325.3 for road junctions* (Table 1).

**Table 1.** Factors for normalizing the fixation durations by area (AOI size).

| Map Feature Type | Areas (Square Pixel) | Inverse Weights |
|---|---|---|
| Green areas | 1,786,107.4 | 17.0 |
| Hydro areas | 939,475.4 | 32.3 |
| Hydro lines | 691,480.1 | 43.9 |
| Roads | 1,481,814.3 | 20.5 |
| Road junctions | 93,290.0 | 325.3 |
| | | |
| *Total Within-AOI* | 4,992,167.3 | 6.1 |
| *Total Outside-AOI* | 30,350,484.7 | 1.0 |

* Within- vs. outside-AOI; weights: if the areas of within- and outside-AOIs add up to 100%.

The average fixation duration of all participants is 315.1 ms for task-relevant (within-AOI) areas, and 288.2 ms for task-irrelevant (outside-AOI) areas of the map stimuli, and the difference in average fixation duration for within- vs. outside-AOI is statistically significant ($F = 125.468$, $p < 0.001$, power = 1.000, see Table 2). Figure 6 demonstrates the normalized differences in fixation duration (ms/area) between within- vs. outside-AOI aggregated by the main variables in the experiment. Figure 6 shows that experts overall have longer fixation durations (top-left), moderate tasks require longer fixation durations on average than easier and harder tasks (top right), and road junctions attract a disproportionate amount of attention from both expert and novice groups (bottom).

**Table 2.** Main effects of independent variables and within- vs. outside AOI analysis.

| Main Effects | Repeated Measures ANOVA | Significance * |
|---|---|---|
| Overall Within- vs. outside-AOI (n = 38) | $F = 125.468$, $p < 0.001$, power = 1.000 | *** |
| Expert (n = 17) vs. novice (n = 21) | $F = 7.610$, $p < 0.01$, power = 1.000 | ** |
| Map feature type (5×) (n = 38) | $F = 23.742$, $p < 0.001$, power = 1.000 | *** |
| Task difficulty (3×) (n = 38) | $F = 55.225$, $p < 0.01$, power = 1.000 | *** |

*Significance: *** $p < 0.001$, ** $p < 0.01$, * $p < 0.05$, $p < 0.10$.*

**Table 3.** Interaction effects of independent variables and within- vs. outside AOI analysis.

| Interactions | Repeated Measures ANOVA | Significance * |
|---|---|---|
| Expertise × AOI type | $F = 0.010$, $p = 0.919$, power = 0.051 | Not significant |
| Map feature type × AOI type | $F = 24.292$, $p < 0.001$, power = 1.000 | *** |
| Task difficulty × AOI type | $F = 11.278$, $p < 0.001$, power = 0.993 | *** |
| Map feature type × Expertise | $F = 3.409$, $p < 0.01$, power = 0.856 | ** |
| Expertise × Task difficulty | $F = 1.780$, $p = 0.169$, power = 0.375 | Not significant |
| Map feature type × Task difficulty | $F = 4.625$, $p < 0.001$, power = 0.998 | *** |
| Expertise × Map feature type × AOI type | $F = 3.743$, $p < 0.01$, power = 0.891 | ** |
| Expertise × Task difficulty × AOI type | $F = 0.241$, $p = 0.786$, power = 0.088 | Not significant |
| Map feature type × Task difficulty × AOI type | $F = 5.154$, $p < 0.001$, power = 0.999 | *** |
| Expertise × Map feature type × Task difficulty | $F = 1.099$, $p = 0.360$, power = 0.522 | Not significant |
| Expertise × Map feature type × Task difficulty × AOI type | $F = 1.288$, $p = 0.245$, power = 0.603 | Not significant |

*Significance: *** $p < 0.001$, ** $p < 0.01$, * $p < 0.05$, $p < 0.10$.*

We present the main effects (inferential statistics) in Table 2, and interactions in Table 3 for the following factors: (i) within- vs. outside-AOI (AOI type), (ii) expertise (expert vs. novice), (iii) map feature type (roads, road junctions, green areas, hydro areas, and hydro lines), and (iv) task difficulty (hard, moderate, easy).

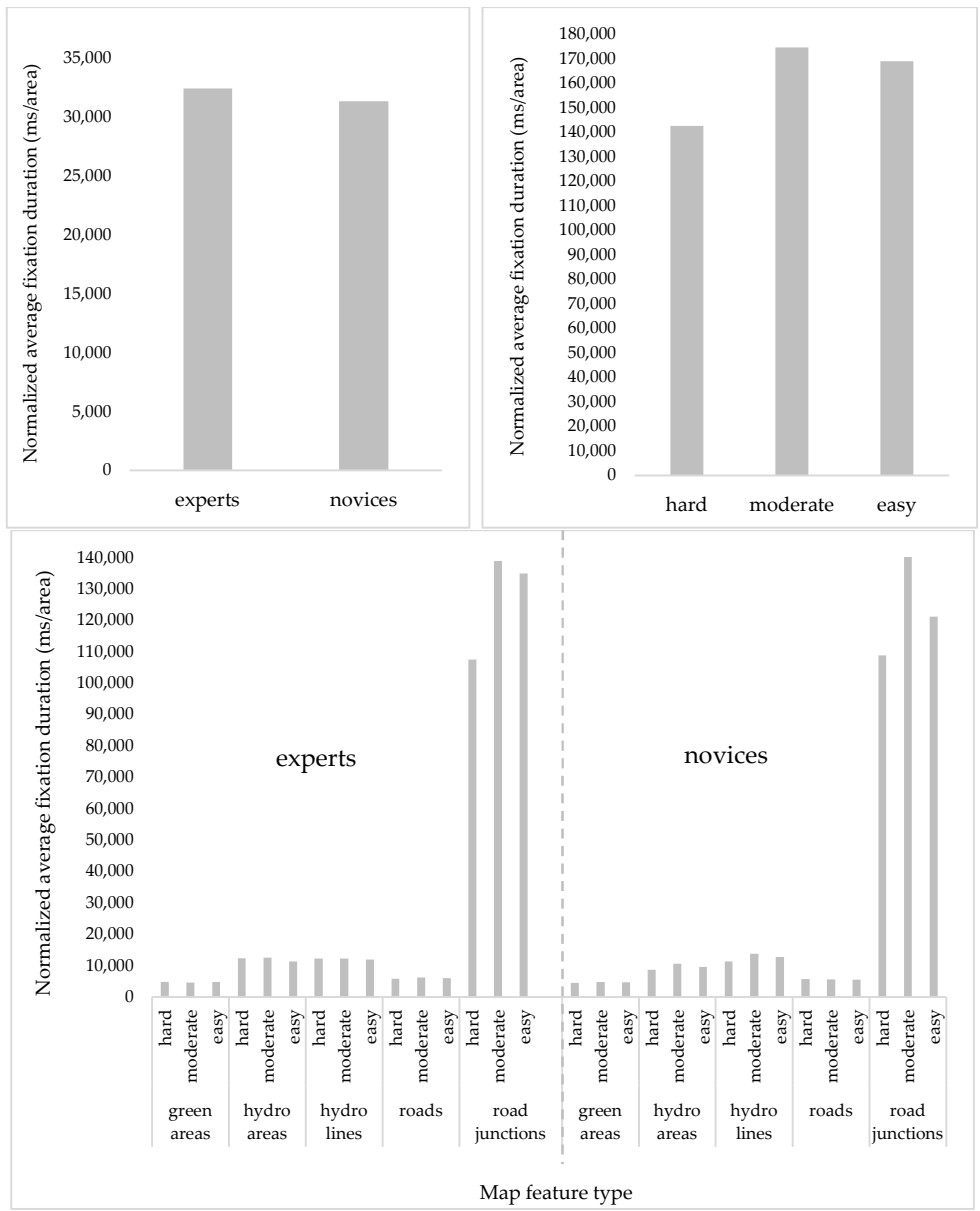

**Figure 6.** Descriptive statistics of normalized differences in average fixation durations (ms/area) within- vs. outside-AOIs. (**Top left**): Aggregated by *expertise*, (**Top right**): Aggregated by *task difficulty*, (**Bottom**): Aggregated by *map feature type*. Inferential analyses are presented in Tables 2 and 3.

Post hoc pairwise comparisons using the *Bonferroni* correction confirmed statistically significant differences in fixation duration. These differences only occurred for task-relevant areas (i.e., within-AOI). Below, we summarize the statistically significant differences:

- **Map feature type**: Road junctions received longer fixation durations than the rest of the map features ($p < 0.001$ ***) followed by hydrographic areas ($p < 0.001$ ***) (Figure 6).
- **Task difficulty**: Moderate tasks received longer fixation durations than easy and moderate tasks ($p < 0.001$ ***).
- **Map feature type × Expertise**: Hydrographic areas ($p < 0.001$ ***) and roads ($p < 0.05$ *) received longer fixation durations from experts.
- **Map feature type × Task difficulty**: Irrespective of task type/difficulty, road junctions received longer fixations than the rest of the map features ($p < 0.001$ ***). For moderate tasks, hydrographic areas received longer fixation durations than other map feature types (hydro areas-green areas: $p < 0.001$ ***; hydro areas-hydro lines:

$p < 0.05$ *; hydro areas-roads: $p < 0.001$ ***) except for road junctions (hydro areas-road junctions: $p < 0.05$ *).

- **Expertise × Map feature type x Task difficulty**: Experts had a significantly longer fixation duration (i) for green areas at hard tasks ($p < 0.05$ *), (ii) for hydrographic areas at moderate ($p < 0.01$ **) and hard tasks ($p < 0.01$ **), and (iii) for roads at moderate tasks ($p < 0.05$ *) (Figure 7).

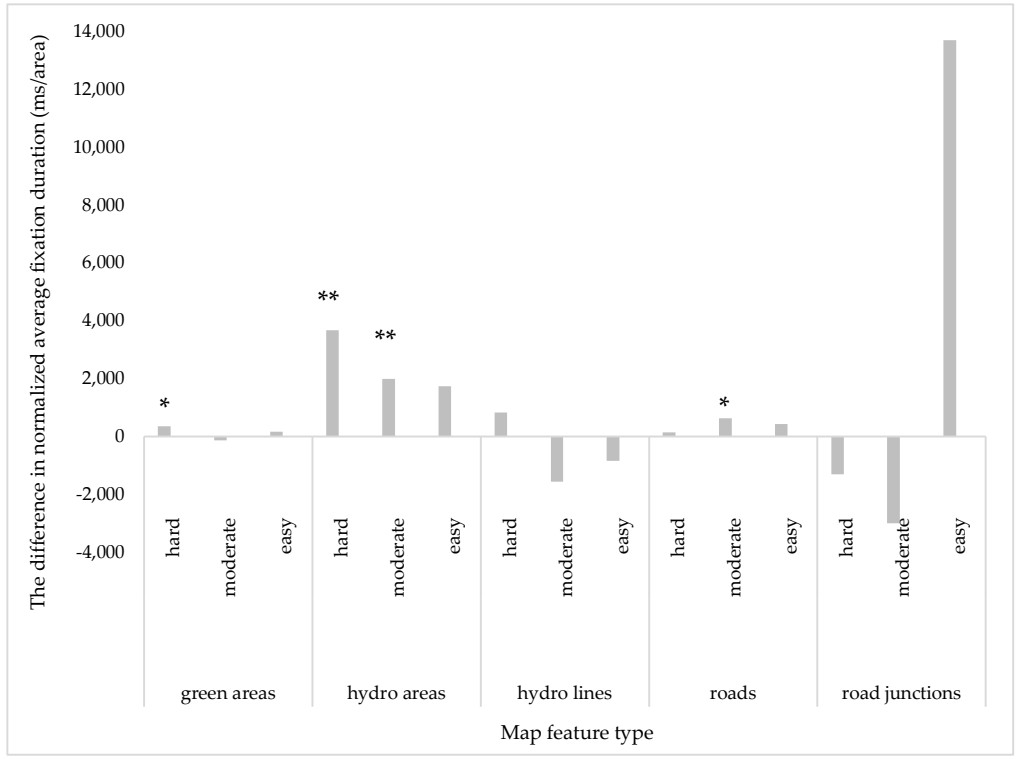

**Figure 7.** The differences in normalized average fixations for experts and novices while they study the task-relevant (within-AOI) areas. For inferential statistics, please see *Expertise × Map feature type × Task difficulty* interaction in the bullet list above. Above zero is more attention by experts, below zero is more attention by novices. *** $p < 0.001$, ** $p < 0.01$, * $p < 0.05$, $p < 0.10$.

*3.2. The Encoding vs. Decoding Stage*

As mentioned earlier, we also conducted a preliminary exploratory analysis visually examining the gaze behavior comparatively between the encoding and decoding stages. We created gaze density maps ('heatmaps') and scan paths for both encoding and decoding stages. Figure 8 demonstrates how experts and novices looked at the same map stimulus (the map in Figure 2), and its corresponding graphical answer screen during Block 1. As a reminder, in Block 1, participants were asked to remember all map landmarks.

The example shown in Figure 8 suggests that experts and novices might have similar visual attention patterns during the decoding stage, for instance, scan paths of both groups show that they were hesitating between options 'a' and 'b' (i.e., the correct answer is 'b' and the difference between 'a' and 'b' is the geometry of the road structure and green areas). Hotspots on the heatmaps during the decoding stage allow us to infer their responses. However, the scan paths of novices look more cluttered (i.e., more transitions between the response options) during the decoding stage. Furthermore, scan paths suggest that novices had a less organized search pattern compared to experts. These patterns persist across many scan paths and gaze density maps we examined visually, leading to the hypothesis that making a decision was likely more difficult or confusing for the novices. A rigorous transition and sequence analysis is beyond the scope of this paper; however, this initial exploratory analysis suggests that these are possible good next steps to further explore.

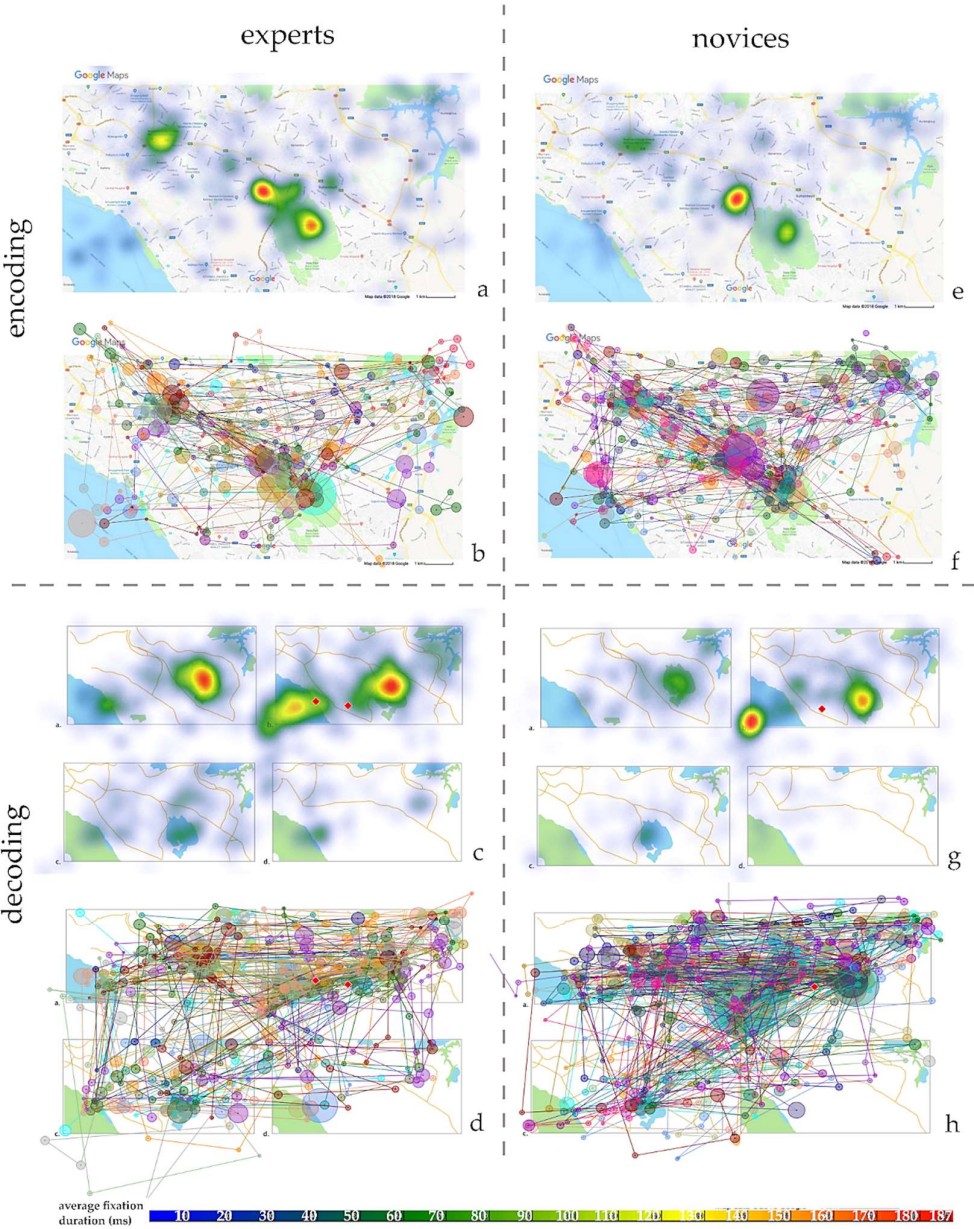

**Figure 8.** Gaze density maps ('heatmaps') and scan paths during encoding and decoding stages for the same map stimulus in Block 1: Original stimulus can be seen in Figure 2. Left: experts ((**a**,**b**): encoding, (**c**,**d**): decoding). Right: novices ((**e**,**f**): encoding, (**g**,**h**) decoding). Heatmaps represent average fixation duration (ms); color codes are shown at the very below. For scan paths, 50 px = 500 ms. (A larger version of the map(s) can be seen in Appendix B.).

*3.3. Participants' Response Accuracy and Response Time*

In addition to eye movement analysis based on fixation durations, we examined participants' response accuracy (success rate) and response (task completion) times. Overall, the experiment resulted in high success rates from both the expert and novice groups and no statistically significant differences emerged between them in terms of response accuracy (presented in detail in [5]). In the scope of this study, we analyzed the success rates by task type and task difficulty, i.e., block by block. We see that all participants achieved the highest success rates for Block 6, i.e., the retrieval of hydrographic features, and the lowest for Block 2, i.e., the retrieval of road and hydrographic features (Table 4).

**Table 4.** Average success rates (%) of experts and novices per task type and task difficulty. Objectively hard tasks marked with red highlight (B1 and B2), moderate in orange/yellow (B3 and B4) and easy ones in green (B5, B6 and B7). Experts score consistently higher but differences are very subtle.

| Task Type | Average Success Rate for Experts (%) | Average Success Rate for Novices (%) | Total Average Success Rate (%) | Total Average Success Rate (%) | Task Difficulty |
|---|---|---|---|---|---|
| B1 (all elements) | 91.5 | 90.9 | 90.8 | | |
| B2 (road and hydro) | 87.3 | 85.6 | 86.8 | 88.8 | HARD |
| B3 (road and green) | 92.1 | 90.4 | 91.3 | | |
| B4 (green and hydro) | 92.7 | 90.6 | 91.5 | 91.4 | MODERATE |
| B5 (green) | 93.5 | 93.0 | 93.3 | | |
| B6 (hydro) | 97.3 | 97.4 | 97.3 | 94.9 | EASY |
| B7 (road) | 94.5 | 93.9 | 94.2 | | |

The overall average task completion (response) time was 4.8 s for experts and 4.0 s for novices, and experts completed all tasks with higher average response times (Figure 9). The difference between the response times of experts and novices was only significant for Blocks 1 (t = 2.636, $p \leq 0.05$), and, Block 2 (t = 2.524, $p \leq 0.05$). In other words, experts took longer than novices in completing the *hard* tasks, but the differences in response time were not statistically significant for the rest of the blocks (Figure 9).

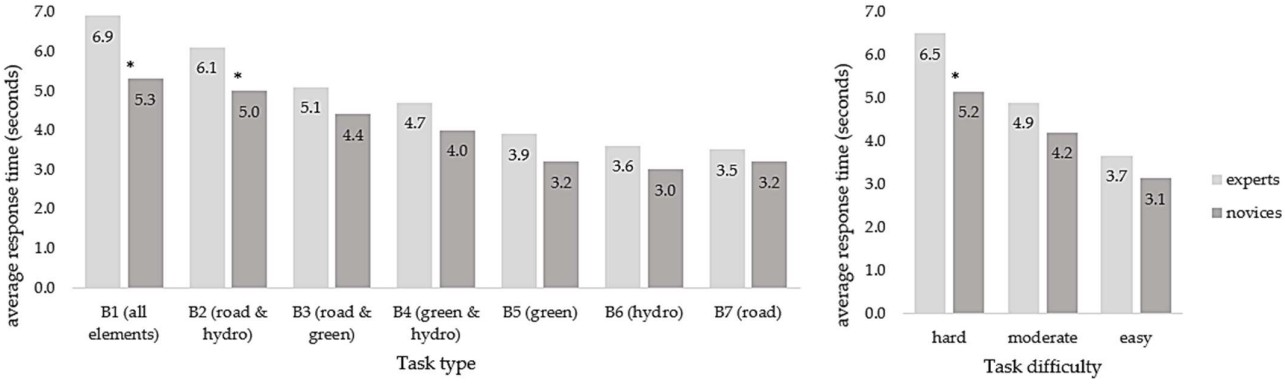

**Figure 9.** Average response times of experts and novices: (**left**): per task type; (**right**): aggregated based on task difficulty. Experts take consistently longer to complete the tasks, but the effect is statistically significant only in some tasks. * $p < 0.05$,.

*3.4. Self-Reported Task Difficulty Rating*

As we have also measured self-reported task difficulty ratings, here we present the outcomes of participants' subjective experience with task groups. Overall, we see that the perceived task difficulty matches the actual task difficulty based on objective performance analysis (Table 5).

**Table 5.** Experts' and novices' self-reported ranking of task difficulty. Results are presented as percentages and each row adds up to 100%. For instance, 48% of the experts thought Block 1 was the hardest. Objectively hard tasks marked with red highlight (B1 and B2), moderate in orange/yellow (B3 and B4) and easy ones in green (B5, B6 and B7). Highest scores are highlighted in gray.

| Task Type | Personal Ranking (Column 1–7: Level of Difficulty in Increasing Order) | | | | | | |
|---|---|---|---|---|---|---|---|
| | 1 | 2 | 3 | 4 | 5 | 6 | 7 |
| B1 (all elements) | 29 | 0 | 5 | 0 | 0 | 19 | 48 |
| B2 (road and hydro) | 0 | 14 | 14 | 14 | 19 | 33 | 5 |
| B3 (road and green) | 0 | 14 | 19 | 29 | 33 | 5 | 0 |

**Table 5.** *Cont.*

| Task Type | Personal Ranking (Column 1–7: Level of Difficulty in Increasing Order) | | | | | | | |
|---|---|---|---|---|---|---|---|---|
| | **1** | **2** | **3** | **4** | **5** | **6** | **7** | |
| B4 (green and hydro) | 0 | 0 | 24 | 38 | 14 | 19 | 5 | |
| B5 (green) | 24 | 29 | 5 | 5 | 5 | 10 | 24 | |
| B6 (hydro) | 29 | 29 | 14 | 0 | 14 | 10 | 5 | |
| B7 (road) | 19 | 14 | 19 | 14 | 14 | 5 | 14 | experts |
| B1 (all elements) | 0 | 6 | 6 | 0 | 6 | 6 | 75 | |
| B2 (road and hydro) | 13 | 0 | 6 | 13 | 19 | 50 | 0 | |
| B3 (road and green) | 6 | 6 | 6 | 41 | 35 | 6 | 0 | |
| B4 (green and hydro) | 7 | 7 | 0 | 27 | 33 | 20 | 7 | |
| B5 (green) | 19 | 19 | 38 | 6 | 0 | 13 | 6 | |
| B6 (hydro) | 31 | 38 | 19 | 6 | 0 | 0 | 6 | |
| B7 (road) | 25 | 25 | 25 | 6 | 6 | 6 | 6 | novices |

## 4. Discussion

In this paper, based on various user-centric metrics and an eye movement analysis, we examined attention patterns of expert and novice map users in a spatial memory (i.e., memorability) experiment using maps that are designed for a general audience (i.e., Google maps). Our approach was similar to Borkin et al.'s [12] approach, as they state that they "measured how visualizations would be remembered if they were images". Similarly, we pursued something that resembles scene memorability. While *expertise* was our main independent variable, because other factors are also known to moderate attention and memorability, we also investigated the influence of *map feature type* and *task type/difficulty* on the memorability of maps. To this end, we conducted an AOI analysis using a self-developed software that automates many steps. Our findings from the experiment show parallels with the existing literature, but also provide new insights. Along with this paper, we release the analysis protocols, as well open a large volume of eye movement data (see Appendix C). In addition to the outcomes from our experiment, we believe releasing our data and protocols will potentially contribute to the generalizability and reproducibility of the findings. We discuss the implications of the findings below by answering our research questions with respect to our hypotheses.

Our results overall confirm our task and map feature type related hypotheses (H1–H4), though with some nuance, and the effects of expertise are somewhat mixed. Not surprisingly, in all conditions, irrespective of expertise, participants spent more time looking at the task-relevant areas (within-AOI), as also demonstrated in previous work [17,44,59], and most statistically significant differences (all after the Bonferroni correction) occur when the participants are attending task-relevant areas (within-AOI). This is interesting and provides further evidence for the well-documented phenomenon since the 1960s that tasks drive eye movement behavior strongly ([60]). It is noteworthy from the perspective of sharing eye movement data that our tasks are not only free viewing; our first task was free viewing "remember what you can", while the others were more goal directed "remember map feature x". Combining eye movement data that are collected in goal-directed (task driven) experiments as well as free viewing conditions would help paint a more complete picture of cognitive processing, i.e., not only how saliency works, but also how cognition affects visual attention.

Our within-AOI vs. outside-AOI analysis provides new evidence, i.e., in this case expertise did not affect task-relevant attention (see AOI type × expertise interaction). We also found out that expertise did not introduce a statistically significant effect on the memorability of map features, while task difficulty and map feature type mattered for remembering the map content. We believe this might be because expert and novice groups

do not differ sufficiently in their exposure to this map type, or the task and stimuli did not require expertise [5]. The fact that we observe some differences based on task difficulty and map feature type but do not observe any differences based on expertise is potentially suggestive, i.e., this finding contradicts with previous evidence that expertise matters in spatial knowledge acquisition tasks [17,42–45], the traditional measures of expertise (education, professional background) may not be a factor if non-experts also have a chance to practice with the map stimuli frequently. If this is confirmed also in future studies, when customizing or personalizing map design for specific groups or individuals, it might be important to measure people's over experience and not only formal expertise.

Moderate tasks received longer fixation durations compared to easy and hard tasks ($p < 0.001$ ***), and a statistically significant difference in response times occurs for hard tasks only. Our pairwise comparisons for fixation durations point out that the effect of *task type* on recognition of map features was statistically significant on *road junctions* and *hydrographic areas*. Irrespective of task difficulty, road junctions received much longer fixations than the rest of the map features ($p < 0.001$ ***) for all participants (i.e., both expert and novice groups). This is followed by *hydrographic areas* ($p < 0.001$ ***) (see Table 2 for interactions). This contradicts our map features x expertise hypothesis (H4) since we observed no pronounced difference between experts and novices. However, it is partially in line with our map feature type × expertise hypothesis (H3); as complex map features (e.g., road junctions) and those large in size (e.g., green areas and hydrographic areas) even if they are simple shapes indeed grabbed more attention, thus appear to be more memorable than moderately complex features. This is possibly mainly due to the known coupling between attention and memorability, and might be potentially explained by visual saliency [50,61]: simpler shapes with large color areas and visually busy areas most likely stand out from their surroundings.

It is important to note that the size of the *road junctions* AOIs (0.3% of total area) was much smaller than that of *hydrographic areas* (2.7% of total area) but they grabbed similar or more of the attention. One way to interpret the prominence of road junctions is that intersections might serve as memory anchors or 'map landmarks', i.e., map readers spend more time looking at those because it helps to orient and remember, e.g., [19,62]. This is a known phenomenon in navigation literature, i.e., people pay attention at the intersections in wayfinding tasks [63]. Similarly, perceptual psychology and computational neuroscience studies provide evidence that sharp edges, such as intersections on a map, are highly salient features [50,64]. Other visual variables (i.e., an object's position, size, shape, value, color hue, orientation, and texture) may also impact visual attention [64]. We considered *road junctions* and *hydrographic areas* both polygon features due to roundabouts and triangle-like shapes the junctions formed; however, evidently they have different characteristics: *road junctions* consist of intersecting yellow lines, generally small but more complex or cluttered, and *hydrographic areas* were mostly larger and depicted using a more subtle color, i.e., light blue. In sum, changes in visual saliency based on different visual variables and visual clutter might be the reasons why large but simple, or very small but distinctive and informative objects attract more attention. This finding is in line with the previous study by Keskin et al. [16]; in which the complexity of the object, e.g., settlements increased the fixation duration. An alternative way to think about why people spend more time looking at road junctions might be that, when two lines meet they form a small point-like feature. Thinking road junctions as point features (rather than polygon features, which we adopted due to enclosed nature of roads and how they segment their surroundings) also invites the possibility that such small-but-important feature require viewers to study the feature longer than other features even if they are larger in size.

Another nuanced observation we have in this study is that *hydrographic areas* ($p < 0.001$ ***) and *roads* ($p < 0.05$ *) received longer fixation durations from experts (H4). The differences in attention between experts and novices were more pronounced in simple polygon features and lines, which are easily accessible in WM and help construct cognitive maps by segmenting the space [5,41,65]. Furthermore, *experts* had a significantly longer fixation

duration (i) for *green areas* at *hard* tasks, (ii) for *hydrographic areas* at *moderate* and *hard* tasks, and (iii) for roads at moderate tasks (Figure 7). While there is not a clear pattern in this specific observation regarding feature types, it is reasonable to speculate that easiest tasks simply did not need much time or attention from either group to generate a difference in gaze behavior, whereas moderate and harder tasks forced experts for a strategy, which may have been to look for map landmarks such as these. Thus, perhaps experts search for map landmarks for help when the tasks get harder while novices may not have such a strategy in place. It is also possible that there are some other moderating effects here though our interaction analyses did not reveal or suggest specific variables.

In addition to fixation duration analysis on task-relevant and task-irrelevant AOIs, we examined two important performance metrics: response time and response accuracy (success rates). Measuring these helps us understand how successfully experts and non-experts remember various map landmarks. Overall, experts outperform the non-experts consistently but very slightly (1–2% difference, statistically not significant) in what appears to be possibly a ceiling effect (all participants are doing very well with all tasks), and they consistently take longer to complete the tasks (Section 3.3., Table 4, Figure 9). Because expertise was necessarily between-subjects (17 experts and 21 non-experts), the statistical power for expertise-based differences is a little lower than for the other variables, nonetheless, the consistency of the descriptive statistics are suggestive and calls for further research. The success rate analysis also verifies the internal validity of our task classification: Harder tasks have the lowest success, and easiest tasks have the highest success rates for both experts and novices.

At the task and map feature level, we see that hydrographic areas received the highest success rates and shortest response times, whereas road junctions had the lowest success rate and the second-longest response times (Table 5, Figure 9). This result was expected as high accuracy scores are associated with selective attention allocation [32]. This is also confirmed by fixation duration analysis in a way that these polygon features are both attentive but some are easier to focus and remember (e.g., hydrographic areas), and some are more complex and require more attention to remember (e.g., road junctions).

As a brief additional exploration, we examined if an implicit confidence difference would emerge between experts and non-experts since there are studies that show such differences based on spatial abilities, which are often correlated to experience and expertise [66,67]. We compared participants' self-reported task difficulty ranking with their performance. Overall, we see that the participants' self-rated task difficulty levels agree with their actual performance, but this is more pronounced for expert participants. In line with the previous literature, this finding suggests that experts may have better metacognition, which further supports the notion that metacognitive skills are trainable. We additionally analyzed participants' map use frequency vs. performance and their usability ranking of Google maps (see Appendix A). However, there is not enough variability among the participants regarding map use frequency (i.e., people have similar levels), or usability ranking, thus no conclusions can be drawn to the effect.

As a qualitative exploration leading us towards future hypotheses, we conducted a preliminary analysis of the eye movements of expert and novice participants during the *decoding stage* through visually (qualitatively) examining the scan path and heatmap visualizations. We observed similar eye movement patterns between expert and novice groups, though the novice participants exhibit more 'chaotic' activity in scanpath visualizations, suggesting that they might be struggling with the decision making more than the experts. Future analyses of response options as AOIs would provide additional insights into participants' attention and visual behavior, i.e., a sequence analysis to see in which order experts and novices examine the answers, and an analysis of transitions between the right vs. wrong answers would give many insights (see Figure 8).

## 5. Conclusions and Future Work

In this empirical controlled laboratory study, we examined expert and novice users' attention and spatial memory strategies while memorizing 2D maps. In the process, we built a technical framework to streamline the AOI analysis of large volumes of eye tracking data.

We assessed the influence of linear and polygonal map features (and implicitly, some visual variables), task type, and expertise on recognition (or cued recall), which is an important component of the overall memorability of a scene. In all conditions, irrespective of expertise, participants spent more time looking at the task-relevant areas and as hypothesized, experts were better at selective attention allocation, therefore ignoring irrelevant areas. However, overall, we observed that task type and map feature type mattered more when remembering the map content compared to expertise. Our brief exploratory analysis of the decoding stage was to raise more future research questions than answers at this stage, and the preliminary analysis indeed indicates more research.

To answer complex cartography questions such as "What kind of visualization/symbolization/ generalization method can be recommended based on the distribution of the map objects?" or "How to control advantaged and disadvantaged map objects equally/in the same way while visualizing, etc.", further analysis and different eye tracking metrics can be considered using our eye tracking data. First of all, a similar AOI-based fixation duration analysis can be conducted for the decoding stage as well to study the retrieval strategies of participants and how they make their decisions. Second, additional eye tracking metrics that are commonly used such as time to first fixation, the number of fixations per second, revisit time, or saccade velocity and length can also be included in both encoding and decoding analysis. More sophisticated eye tracking metrics can be calculated to gain a better insight into the attentional patterns of map users over the execution of the task. For example, K-coefficient [68], which is the difference between fixation duration and its subsequent saccade amplitude, indicates whether visual behavior is focal or ambient. Similar to Krejtz et al. [69], K-coefficient can be used to study focal vs. global attention patterns between experts and non-experts over the duration of the task. On the other hand, gaze transitions from one AOI to another can be traced by calculating transition frequency which corresponds to the number of times a reviewer transits fixations between two paired-AOIs [70].

The large-scale eye tracking dataset which is described in the present work could be also utilized for the generation of statistical grayscale heatmaps (referred also as "grayscale statistical heatmaps" in the international literature) as well as for the computation of statistical indices based on them. A statistical grayscale heatmap is a quantitative product that is generated based on the use of point data distributions that are referred either to raw gaze data or to fixation points' centers (see [71] for more details). Moreover, statistical grayscale heatmaps could also serve as an aggregated gaze data visualization method since they can represent the overall spatial allocation of visual attention of multiple observers. Several eye-tracking datasets (e.g., EyeTrackUAV [55], and EyeTrackUAV2 [54]) include such products aiming at the distribution of an objective ground truth that could feed deep learning approaches (see, e.g., the recent study provided by Gökstorp and Breckon [72]). Furthermore, recently Cybulski and Krassanakis [73] used statistical grayscale heatmaps for the development and the computation of a series of statistical indices that aim to indicate the visual strategies of map users during the execution of a specific target-based cartographic task. More specifically, aiming at the examination of the effect of map label language on visual search, they proposed five indices which were used in order to indicate the spatial allocation of map readers' visual attention in the center and in the periphery of different cartographic backgrounds during searching for point symbols with labels in Polish and Chinese [73].

The free distribution of eye tracking datasets produced during the observation of cartographic backgrounds (see, e.g., the dataset distributed by Tzelepis et al. [74]) could substantially help towards modeling visual attention by developing dedicated saliency models. Such models have been developed in order to predict visual attention during the

observation of natural images. However, a map is an artificial and abstract image that aims to represent a part of the real world. Hence, typical saliency models fail to predict map user behavior (see, e.g., the work provided by Krassanakis [75]). Broadly speaking, eye tracking provides valuable information in studying spatial cognition [6,76], and we will continue with further analytical and methodological research in this area. Questions such as the following will guide our future efforts: Can we predict and model the behavior of map readers by utilizing machine learning and predictive models applied to the large volumes of already collected eye tracking data (possibly in combination with other measures where available), and adapt the map to user needs in a personalized manner in real time?

**Author Contributions:** Conceptualization and methodology, Merve Keskin, Arzu Çöltekin and Vassilios Krassanakis; software and validation, Vassilios Krassanakis; formal analysis, investigation, resources, and data curation, Merve Keskin; writing—original draft preparation, Merve Keskin; writing—review and editing, Merve Keskin, Arzu Çöltekin and Vassilios Krassanakis; visualization, Merve Keskin. All authors have read and agreed to the published version of the manuscript.

**Funding:** This research received no external funding.

**Data Availability Statement:** We make the dataset available within the publication of this article. Please see Appendix C for the description of the dataset.

**Conflicts of Interest:** The authors declare no conflict of interest.

## Appendix A. The Post-Test Questionnaire

**Table A1.** User characteristics of the recruited participants [48].

| % | Q1: Please Choose the Highest Level of Education You Have Completed | | Q2: How Often Do You Use Google Maps? | | Q3: On a Scale of 1-5, with 5 Being "Strongly Agree" and & Being "Strongly Disagree" Please Answer: Do You Think Google Maps is Easy to Use? | | Q4: What Do You Think about the Experiment? | |
|---|---|---|---|---|---|---|---|---|
| | | *N* | | *N* | | *N* | | *N* |
| **Experts (N = 17)** | PhD | 1 | everyday | 10 | 5 | 13 | Positive | 11 |
| | MSc | 16 | once/twice a week | 6 | 4 | 4 | Neutral | 2 |
| | | | once a month | 1 | 3≤ | 0 | Negative | 4 |
| **Novices (N = 21)** | MSc | 11 | everyday | 8 | 5 | 8 | Positive | 5 |
| | BSc | 8 | once/twice a week | 11 | 4 | 11 | Neutral | 10 |
| | High School | 2 | once a month | 2 | 3 | 1 | Negative | 6 |

## Appendix B. Larger Versions of Maps in Figure

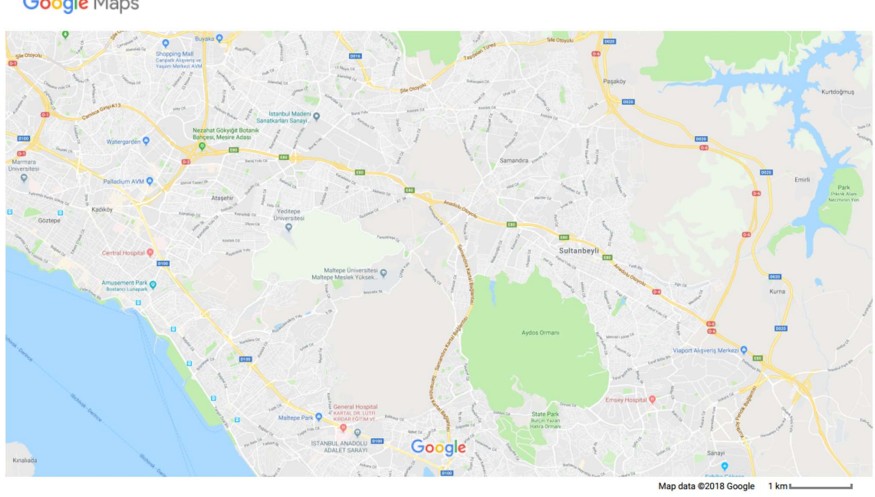

**Figure A1.** Larger version of Figure 2a.

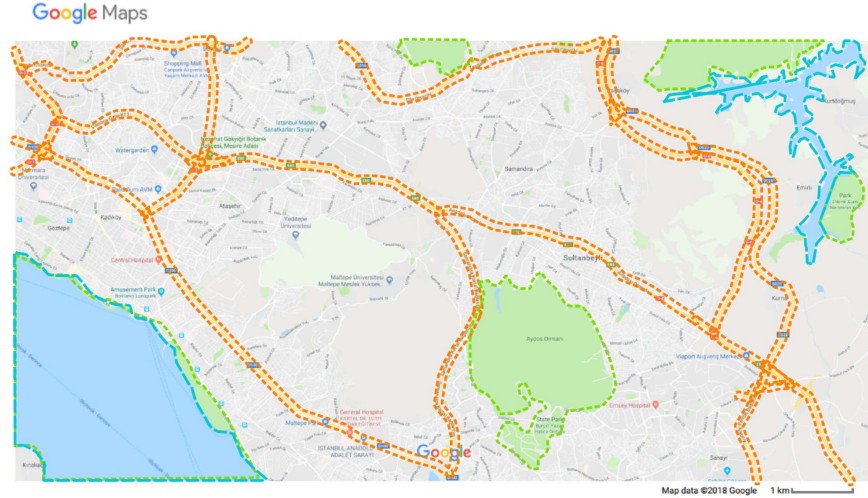

**Figure A2.** Larger version of Figure 4.

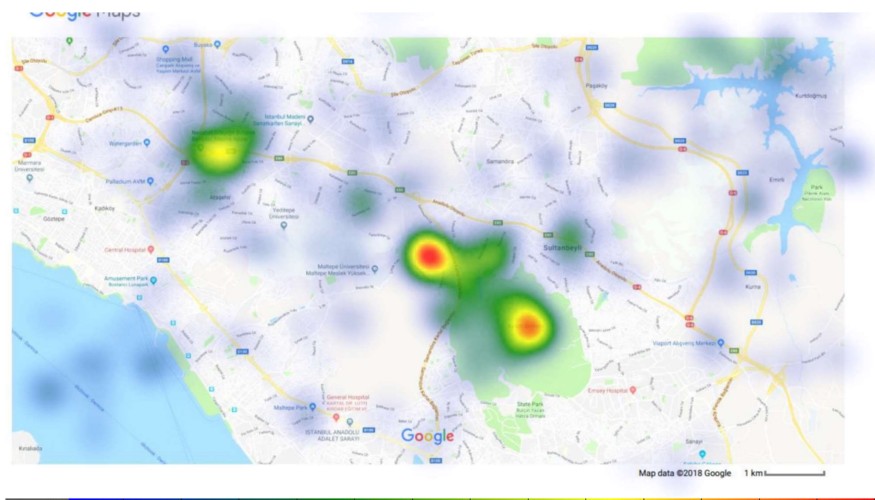

**Figure A3.** Larger version of Figure 8a.

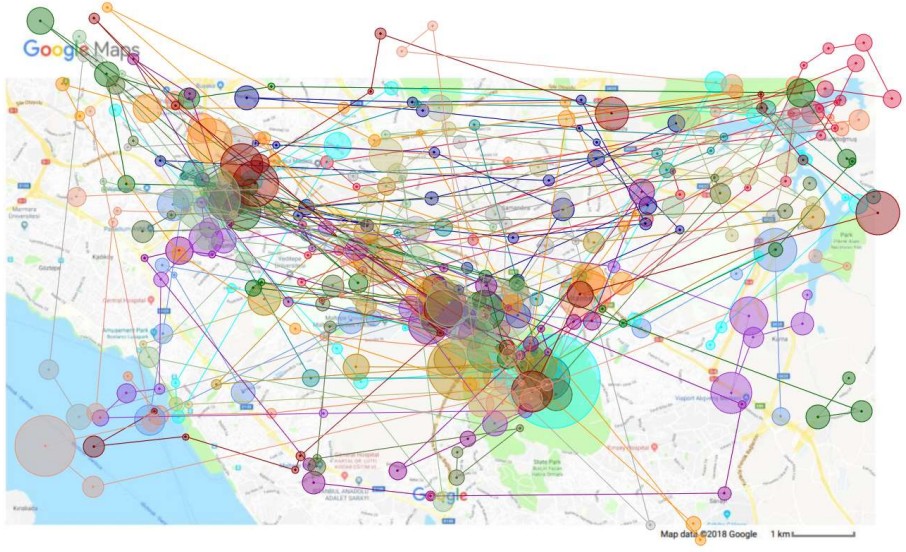

**Figure A4.** Larger version of Figure 8b.

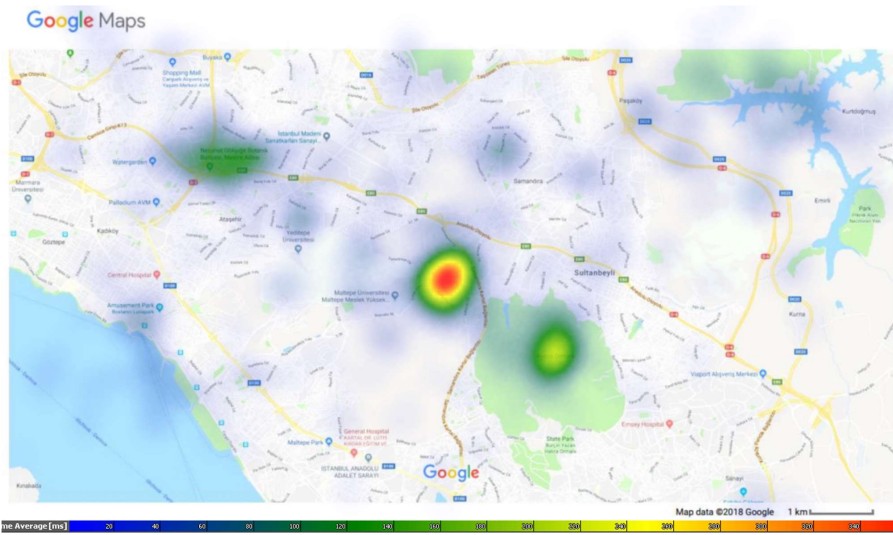

**Figure A5.** Larger version of Figure 8e.

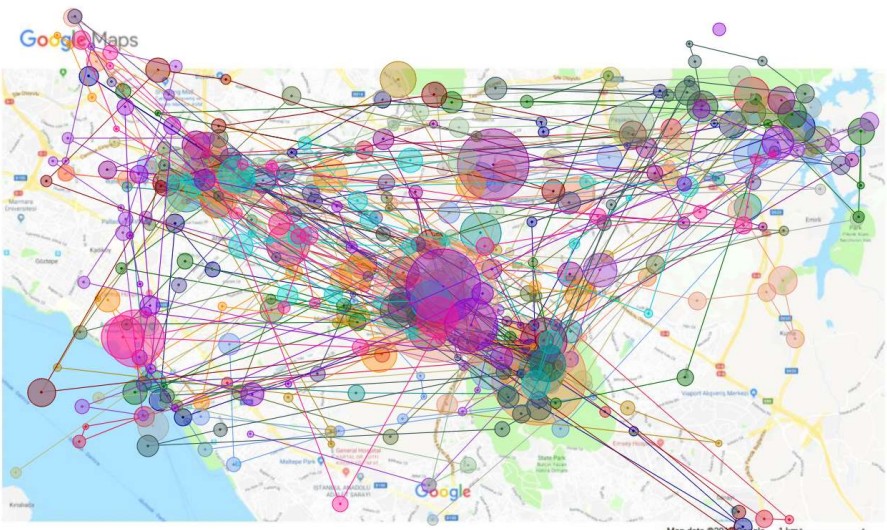

**Figure A6.** Larger version of Figure 8f.

## Appendix C. The Description of the Dataset

CartoGAZE constitutes a large eye movement dataset produced during the observation of cartographic products. The production of this dataset was based on the collection of gaze data (SMI RED250 eye tracker) during the observation of the snapshots of 2D static Google navigational maps under free-viewing conditions. The dataset includes the map stimuli, the AOI files, the task descriptions, and full procedural details for the reproducibility of results and to create possibilities for future research.

We aim to develop an automatic and semi-automatic AOI-based analysis model capable of detecting attentive areas from the point of view of human operators, by considering the vector characteristics and visual variables of the map features, task difficulty, expertise, and spatial memory strategies of human operators.

CartoGAZE is freely distributed to the scientific community via Harvard Dataverse: https://doi.org/10.7910/DVN/ONIAZI (accessed on 1 December 2022).

The raw data was collected in the framework of the doctoral research project entitled "Exploring the Cognitive processes of Map Users Employing Eye Tracking and EEG".

There are six folders in the data repository; Raw_ET_data, AOIs, Map_stimuli, Final_Fixations, AOIs_all_results, and Calculate_AOI_areas folders which are explained below:

1. Raw_ET_data: The collected eye tracking data in txt format, and can be linked with the map stimuli using the SYNC number in lines with, e.g., # Message: SYNC 103.
2. AOIs: The coordinates of AOIs are shared as csv file format.
3. Map stimuli: The snapshots of 2D static Google navigational maps were taken at zoom level 15 and approx. 1:40 k scale. The total number of map stimuli used in the experiment is 37 and can be found under Map_stimuli folder in png format. Below are some examples (*see* Table A2 for the resolution of the stimuli) (Figure A7):

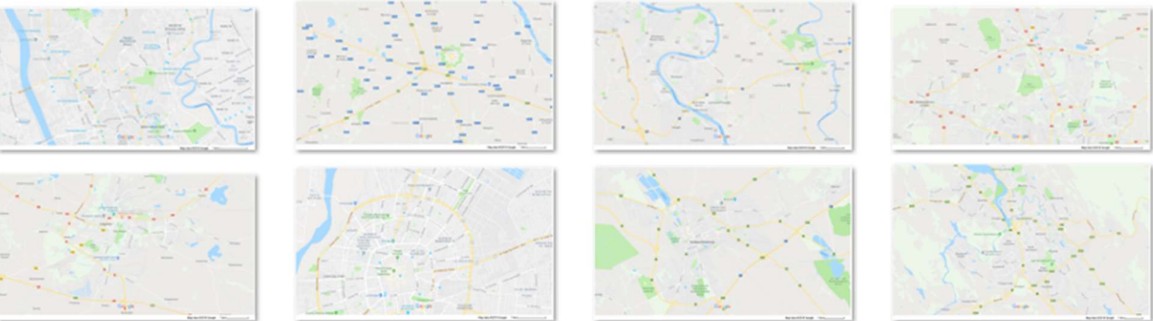

**Figure A7. Examples of map** stimuli in CartoGAZE dataset.

4. The outputs of the scripts can be found in Final Fixations and AOIs_all_results folders as single. csv files organized per stimulus, map feature type, participant and task type. The explanation of columns are as follows:

- Column 1: Gaze position X (left eye),
- Column 1: Gaze position Y (left eye),
- Column 1: Trial duration,
- Column 1: Duration start,
- Column 1: Duration end, and
- Column 1: The number of points clustered in the same fixation.

In AOI files, the rest of the columns after the above-mentioned six columns represent within- or outside-AOIs: If the summary of the rest of the columns $\leq 1$, the fixation is inside the AOI, else outside AOI (represented as 0s and 1s). These files under AOIs_all_results are also aggregated in one. csv file named "AOI_all_results".

5. Calculate_AOIs_areas: includes the areas of AOIs (square pixel) separately for each stimulus and map feature type, and aggregated in a single file named "all_AOI_areas.csv".

**Table A2. The resolutions of the stimuli**.

| Stimulus ID | Width | Height | Stimulus ID | Width | Height | Stimulus ID | Width | Height | Stimulus ID | Width | Height |
|---|---|---|---|---|---|---|---|---|---|---|---|
| **101** | 1148 | 660 | **131** | 1356 | 773 | **153** | 1199 | 587 | **173** | 1338 | 759 |
| **103** | 1148 | 660 | **133** | 1360 | 752 | **155** | 1418 | 834 | **175** | 1218 | 692 |
| **105** | 1721 | 877 | **135** | 1353 | 787 | **157** | 1336 | 756 | **177** | 1246 | 733 |
| **107** | 1210 | 613 | **137** | 1354 | 792 | **159** | 1419 | 834 | **179** | 1124 | 669 |
| **117** | 1279 | 660 | **139** | 1354 | 787 | **161** | 1198 | 591 | **181** | 1336 | 803 |
| **119** | 1356 | 785 | **141** | 1326 | 750 | **163** | 1196 | 588 | **187** | 1207 | 591 |
| **121** | 1357 | 778 | **145** | 1248 | 735 | **165** | 1338 | 758 | **189** | 1211 | 596 |
| **123** | 1360 | 776 | **147** | 1336 | 805 | **167** | 1342 | 755 | **191** | 1129 | 717 |
| **127** | 1292 | 708 | **149** | 1337 | 756 | **169** | 1222 | 720 | | | |
| **129** | 1281 | 698 | **151** | 1340 | 809 | **171** | 1245 | 731 | | | |

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
