# Peer review of "Visual Attention and Recognition Differences Based on Expertise in a Map Reading and Memorability Study"

_ijgi, doi:10.3390/ijgi12010021_

Round 1

Reviewer 1 Report (Previous Reviewer 1)

The authors have addressed most of my critiques in their revised manuscript. There are a few small issues with some of the revised material.

p. 8, line 3: substitute recognition for recall

Figure 6 caption could be clearer. Which is on the left or the right bottom — expert or novice?

Page 14: ‘the only difference between ‘a’ and ‘b’ is the geometry of the road structure. —> not from what I can observe. There are green areas present in ‘b’ the are not present In ‘a’.

Page 17: The fact that we do have some differences based on expertise but do not find a difference based on expertise is a valuable observation (i.e., counterintuitively in some cases the ‘traditional’ measures

of expertise may not be a factor).

Sentence does not make sense. You have some differences but don’t find differences? Be more clear and specific about what you are talking about here. I think you mean you do not find a memorability difference based on expertise, but you do find differences in gaze behavior?

Page 17: statistically significant difference in response times also occurs for hard tasks only

Delete ‘also’ - you have modified the previous sentence so also does not make sense anymore.

In addition, there is mention of recall here, but you’ve changed it to recognition elsewhere. The whole manuscript should be carefully searched to see if you’ve missed other instances of this.

Page 17: One way to interpret the prominence of road junctions is that intersections might serve as memory anchors or ‘map landmarks’, i.e., map readers spend more time looking at those because it helps to orient and remember.

Perhaps this is also consistent with the RU Bochum studies of map grid lines and memory.

Response to Minor Point 3: I did understand that a fixation cross was shown between trials. What I was asking about was whether you considered a blank screen between presentation of the map stimulus (7 seconds) and displaying the four skeletons.

Author Response

Please see the pdf attached.

Reviewer 2 Report (Previous Reviewer 2)

Changing the "recall" keyword to "recognition" is correct. However, supplementing a long text with a lot of additional data with next considerations makes it even more difficult to focus on following the research. An example is the descriptively extended conclusions, instead of listing the most important new conclusions compared to other publications. The maps in the experiment are still not readable.
The article can only be understood by a very close group of eye-tracking specialists, because it deals in detail with the techniques of this method. The authors have not shown permission from GoogleMaps to publish portions of this map.

Author Response

Please see the pdf attached.

Reviewer 3 Report (New Reviewer)

This paper details the results of an empirical study investigating the memorability and recall of map landmarks between experts and novice map users. The findings indicate better selective attention allocation by experts. However, expertise is not the decisive factor when map content is to be remembered; rather task type and map feature type matters more is such respect. The manuscript soundly structured and related work is adequately referred.

Although I have been requested to join the review process at a second stage, and I am able to see the additions/changes/corrections of the authors to the original manuscript, I still feel some of the major concerns of the first round reviewers are not successful deal with.

a) How are map designers (cartographers) to make use of these findings? – The authors have added a paragraph at the end of the Introduction Section but I do not think that what is stated therein in convincing enough. Memorability and recall are very important factors in any visual scene, and for 2d maps having the users remembering map landmarks is an indicator of success of the map itself. Nonetheless, I do not think measuring how many a user can remember out of a task context e.g. (a) going from A to B, having to identify decision points and developing a navigation strategy, or (b) identifying specific map features etc., adds to the discussion of memorability since other map features may be salient to different task-driven scenarios. It would be more interesting to see which of the map landmarks are noticed first (which features subjects gaze at first) and try to infer why based on the landmarks’ map placement, visual variable(s) and so on.

Finally, regarding the motivation added segment I think it should be placed right after the three bullet-listed research questions and before the literature discussion.

b) Are the findings really groundbreaking in any respect? – I believe not.

·       Regarding why participants were paying more attention to road junctions the authors have answered by adopting the first round reviewer #1 suggestion that junctions are usually used in navigation decision making and this also an evidence that we should erase the task-driven approach from this kind of survey. Moreover, I would like to add another attribute of road junctions that may require more time spent on them; their small size and this also has to do with the feature per se. With this in mind, road junctions should be better considered as points instead…

·       Regarding why expertise is not a decisive factor as highlighted by the results, it has also been pointed out by the previous round reviewers that Google Maps is nowadays on every mobile device thus, it should be expected that expertise would not be able to differentiate any of the results, especially since this is a “task-neutral” survey. Therefore, no expertise is really needed or “mobilized” to perform any of the survey tasks requested from the sample, and this is also verified by the results in accuracy for both groups.

·       By the way, subjects’ accuracy diminishes as difficulty increases constitutes a finding that is also expected and does not provide any useful insight.

Apart from the importance of the findings as such, I am also concerned about fatigue and learning effect that might have some nuance in the findings. Since 37 map excerpts were shown to participants in each block and they are all from Google Maps, showing semi-rural regions I do not think that you have avoided learning effect even if fatigue had been prevented as you mention. They same kind of map will more or less the same content, with the same visual variable, styles, lettering, more or less of the scale and so on, I would not exclude learning effect from the picture…

Finally, I must say that this new version clarifies things much better than the previous one, thanks to the meticulous review process of round 1 and the will of the reviewers to tackle these issues. However, I still believe that the methodological and research concerns of the reviewers have not been adequately addressed due to the design of the survey in the first place. Moreover, there are some minor typos, expression errors in the added sections  that need attention.

Author Response

Please see the pdf attached.

Reviewer 4 Report (New Reviewer)

As I am not qualified to interpret studies of eye movement/fixations, I cannot say if this is better than what exists in the literature. My main reservation is that it provides little if any practical guidance to cartographic and UI design. That so little difference was found between expert and novice map users indicates to me that little of practical use was learned. The conclusion states:

To answer complex cartography questions such as “What kind of visualization / symbolization / generalization method can be recommended based on the distribution of the map objects?” or “How to control advantaged and disadvantaged map objects equally/in the same way while visualizing, etc.”, further analysis and different eye tracking metrics can be considered using our eye tracking data.

My own conclusion is "why bother?". No matter how conceived and executed, I feel eye tracking will always yield equivocal results. If the authors' goal is to improve the communication value of maps, perhaps they should focus on the most widely used types of maps with alternative methodologies; in my opinion psycho-physical approaches don't seem to help to understand higher-level (geo)spatial cognition.

It is sound work, carefully executed and reported, and thus deserving of publication. Some professionals may find it of heuristic value. I did not.

Author Response

Please see the pdf attached.

This manuscript is a resubmission of an earlier submission. The following is a list of the peer review reports and author responses from that submission.

Round 1

Reviewer 1 Report

This manuscript presents the results of an empirical experiment investigating the memorability of map landmarks. In some ways, the experiment was well designed, but in others, less so. The manuscript itself is logically structured and written in good idiomatic English, and it makes reference to appropriate related work.

Major concerns
1. I have concerns with the way that expertise is operationalized in this experiment. I wouldn’t necessarily expect people who are not geospatial professionals to have limited experience with this type of map. Given that they are in every smartphone and that most people would use these maps frequently (at least in Europe where the work was conducted and where the quality of underlying map data is high) =, it’s possible that there is no relevant actual expertise difference whatsoever between the two groups. It would have been better to ask people detailed questions about their use of and experience with Google Maps and other similar maps and from that construct some expertise levels.

2. The task I think also could have had better ecological validity. If I’m using such a map and need to remember a landmark, it’s usually because it’s along a route that I intend to travel, not because I am wanting to know where all road intersections or all hydrographic features on the screen are. The memorability of a single feature may be quite different to remembering all features. Some discussion of the validity of the task should be added to the manuscript. I wonder also if the reason why participants were paying more attention to road junctions is because these are environmental features usually used in navigation decision making, particularly at the scale at which the maps were delivered.

3. Is recognizing something the same as memorability? I think there’s a distinction between recognition and recall, and it seems to me that you are measuring recognition rather than recall. This point should be discussed in more depth and the relationship of recognition and recall to memorability should be clearly outlined. Does one have to be able to recall features for them to be memorable? If so, then your experiment hasn’t really measured this, in my view. A better test of recall would be to identify where a feature that is not drawn on the map is located, either by clicking or sketching.

4. Shouldn’t the relevant/irrelevant locations vary by block? It seems you’ve used the same set of relevant locations no matter which block you’ve analyzed.

Minor points:

  1. It’s not really clear how the blocks were delivered. Sequentially? Randomized? Also, Figure 1 seems to suggest there were 50 trials per block, not 37 as described in the text. With a large number of trials are there fatigue effects in the experiment? Do I understand correctly that the same 37 locations are used in each block? Could there be a learning effect of the locations over time? Is this controlled for in the delivery of the blocks?
  2. It would be useful for Figure 2 to include all of the possible answer options so that the differences (and therefore the difficulty in recognizing the correct one) are easier to understand. I’d also like to know a bit more about how the distractor options were constructed.
  3. Why 7 seconds for the presentation time? Did you consider a blank screen between the map stimulus and the presentation of the possible responses (or another fixation cross)?
  4. It’s not clear to me why the normalization was done with the Outside-AOI instead of the total screen area.
  5. Should Table 3 be labeled Interaction Effects?
  6. Bulleted list on p. 11: seems like expertise x map feature type is listed twice, with different findings reported.
  7. Bulleted list on p. 11: you note that hard tasks have longer fixation durations, but that’s not what Figure 6 shows. I’m confused.
  8. Line 449: do you mean decoding instead of encoding?
  9. Figure 8: from looking at the scan paths, it looks like novices had a greater number of long duration fixations. This seems at odds with the aggregated patterns. Is there something unusual about this particular trial?
  10. I don’t quite understand how the percentages in Table 5a and 5b were calculated. Shouldn’t the rows total to 100%? They seem to total to 123% in 5a and 76% in 5b.

Reviewer 2 Report

The proposed article meets the requirements of a scientific article. However, there are a few things that need to be clarified. The keywords are missing the type of map: Google Map and in the text a description of the features of this map design. Why was there no question for participants: how often do they use Google Maps? It can be assumed, after all, that even novices for geomatics and GIS who are daily users of Google Maps - will be experts for this map.
The Googe Map is not in Mercator projection but in WEB-Mercator projection
The quality of the figures should be better: descriptions and maps are illegible, eg figures 7 and 8. The description of figures 2-4 lacks the Google Maps province.

Reviewer 3 Report

This paper is not motivated.   Why is the memorization of map features in an interactive interface such as Google Maps necessary?   Why not rely on recognition instead of recall?  Also, what would someone do with these results?  There doesn't seems to be any significant insights not already known.    Its also not clear what the value of the released data or their self-described framework is.   It leaves it up to the readers imagination.

Other aspects of concern:

- its not clear how the 3 levels of difficulty were quantitatively judged

- its not clear if colour mattered or if the participants were screened for colour blindness

- is there any literature support for the kind of AOI area normalization that was used?  in particular, the 325.3 normalization factor is enormous.  Did that alone create the map feature results in figure 6?

- lines 395 and 419 seem to contradict each other with regards to what requires longer fixation durations (moderate or hard?)

In terms of presentation and writing, it is generally well done.  Except:

- that the large paragraph on page 3 is too long and the conclusions seemed too long as well.

- explain what you mean by the encoding and decoding stages

- explain what a mixed factorial block user study design is and how it works, referring to and explaining figure 1

Reviewer 4 Report

Maps are essential to our everyday lives, so this kind of research is very valuable and desirable, even though it has been around for many years and it is difficult to present something new as a new result. The most valuable part is contained in the question of how do experts and novices differ in terms of recall performance (expertise), because this has not been investigated before (Figure 6). We definitely recommend the continuation of research into cartographic visualization and recommendations that will be the conclusions after the research of what kind of visualization / symbolization / generalization method can be recommended based on the distribution of the map objects?